# Dietary Inclusion of Mushroom (*Flammulina velutipes*) Stem Waste on Growth Performance, Antibody Response, Immune Status, and Serum Cholesterol in Broiler Chickens

**DOI:** 10.3390/ani9090692

**Published:** 2019-09-17

**Authors:** Shad Mahfuz, Tengfei He, Sujie Liu, Di Wu, Shenfei Long, Xiangshu Piao

**Affiliations:** State Key laboratory of Animal Nutrition, College of Animal Science and Technology, China Agricultural University, Beijing 100193, China; shadmahfuz@yahoo.com (S.M.); hetengfei@cau.edu.cn (T.H.); heiluobo12300@gmail.com (S.L.); superwudee@163.com (D.W.); longshenfei@cau.edu.cn (S.L.)

**Keywords:** antibody response, broilers, growth performance, lipid metabolism, mushroom stem waste

## Abstract

**Simple Summary:**

The continued overuse of antibiotics in the poultry industry with the purpose of increasing production performance and health status has led to human health hazards. This research explores the use of medicinal mushrooms to get rid of antibiotics in poultry feed without affecting optimum performance. Most medicinal mushrooms contain biologically active substances such as polysaccharides, glycoproteins, and other macromolecules, which can serve as good dietary supplements and immuno-modulating agent in chickens. Therefore, the objective of this study was to examine the effect of *Flammulina velutipes* mushroom stem waste (MW) on Growth performance, antibody response, immune status, and serum cholesterol in broiler chickens.

**Abstract:**

This study was carried out to investigate the effects of mushroom (*Flammulina velutipes*) stem waste (MW) on growth performance, antibody response, immune status, and serum cholesterol in broiler chickens. A total of 252 1 day old Arbor Acres (AA) male broiler chicks were randomly assigned into four treatments with seven replications of nine chicks each. The duration of experimental period was total 42 days. Dietary treatments includes a standard basal diet as negative control (NC) group; control diet with antibiotics (Chlortetracycline) considered as positive control (PC) group; 1% mushroom stem waste (MW) fed group; and 2% MW fed group. No significant differences (*p* > 0.05) was observed on average daily feed intake, body weight gain, and feed conversion ratio among experimental groups. Antibody titers against Newcastle disease (ND) and infectious bursal disease (IBD) were higher (*p* < 0.05) in 2% MW fed group than NC and PC fed groups. Serum immunoglobulin G (IgG) was higher (*p* < 0.05) in both levels of MW fed groups than in the NC and PC. Serum interleukin-2 (IL-2), interleukin-4 (IL-4), interleukin-6 (IL-6), were higher (*p* < 0.05) in 2% MW fed groups than in the NC and PC fed groups. Total cholesterol concentration was lower (*p* < 0.05) in both levels MW fed groups than in the NC. High density lipoprotein cholesterol (HDL) was lower (*p* < 0.05) in both levels of MW fed groups than that of NC and PC fed groups. MW at 2% level can be used as potential phytogenic feed supplement in broilers.

## 1. Introduction

Traditionally, mushrooms have been grown for human food as well as pharmacological purposes in different countries. The beneficial properties of mushrooms’ bioactive compounds indicated their potentiality to be used as performance-enhancing natural feed additives for livestock [1]. Antibiotics have been used in broilers production for growth promoters as well for the prevention of bacterial infections since many years. However, the continuous and over application of in-feed antibiotics in poultry production has created public health hazards [2]. Movahhedkhah et al. [3] reported that application of natural herbs and their extracts has growth-enhancing roles in chickens. Due to huge economic losses by gastrointestinal infections and of strict laws to apply in-feed antibiotics, there is a demand for a natural alternative therapeutic agent [4]. Mushrooms havelong been popular for their health benefits and medicinal and tonic attributes [5,6]. *Flammulina velutipes* is a very common edible mushroom. It is known as the winter mushroom, needle mushroom or enoki mushroom and has worldwide distribution [7,8]. *F. velutipes* mushrooms arean excellent source of protein, vitamins, minerals, and unsaturated fatty acids [8]. In addition, it has been reported as an immune modulatory effect via stimulating immune response, production of cytokines, and antibacterial, antiviral, antifungal, antioxidant, and anticancer activities [9,10]. *F. velutipes* mushroom hold the phenolic component with the higher antioxidant and immune activities [11]. Higher market demand has led the increased production of mushroom stem base, which is treated as a waste material in the environment, but its utilization is still limited [12]. Stem base is the stem waste of *F. velutipes* mushroom. *Flammulina velutipes* stembase is the waste left over after the edible part of *F. velutipes* is harvested. China alone produces more than 100,000 tons of *Flammulina velutipes* stem waste each year, and the production is even higher in developed regions, such as Europe, America, Japan, and South Korea [13]. Currently, the stem waste of *F. velutipes* is used as compost and the majority of the stem base is not properly utilized, which is wasteful [14]. At present, higher attention has been rewarded to the efficient utilization of agricultural residues, with a view to reducing production costs and alleviating the environmental pollution caused by the residues [15]. Newcastle disease (ND) and infectious bursal disease (IBD) are very common viral infections in broilers, which cause huge economic losses [16]. There are no effective treatments; vaccination is the only way to prevent birds from those viral diseases. Both diseases may affect feed conversion ratio (FCR) as well as delay market weight gain. Modern poultry producers want to keep the flock free from diseases by improving vaccines techniques, instead ofusing antibiotics drugs. Thus, the research for effective, environmentally pleasant, and secure feed additives have become essential in poultry production systems [17].

To our knowledge, very limited studies have been carried out yet to examine the efficiency of mushroom waste on performance, and health status in broilers. The objective of this study was to evaluate the possibility of disposed mushroom (*Flammulina velutipes*) stem waste (MW) as natural feed supplement on growth performance, antibody response, immune status, and serum cholesterol in broilers.

## 2. Materials and Methods

### 2.1. Experimental Design, Chickens, and Dietary Treatment

The experiment was carried out at the animal shed building under the college of Animal science and Technology, China Agricultural University. A total of 252 1 day old Arbor Acres (AA) male broiler chicks were randomly assigned into 4 equal treatment groups, with seven replications of nine chicks for each treatment (63 chicks per group). All chicks were numbered individually by wing tag and were weighed prior to place into different replicated groups. The average initial body weight was 45.16 ± 1.07g (mean ± std). Chickens were given a standard basal diet considered as negative control (NC) group; control diet with antibiotics (Chlortetracycline) considered as positive control (PC) group; 1% mushroom stem waste (MW) fed group; and 2% MW fed group, respectively. Feed and water were offered ad libitum during the whole experimental period of 42 days. Care and management of the experimental broilers were approved by the Animal Careand Use Committee of China Agricultural University (Beijing, China).

The collected mushroom stem waste (MW) was dried and mixed with experimental diets according to the National Research Council [18] standard. The starter diet was supplied from 1 day to 21 days and the finisher diet was supplied from day 22 to day 42 days. Chicks were reared in a temperature-controlled room, and had free access to clean water and mash feed in the three-layer wired cages (120 cm–60 cm–50 cm). The experimental house temperature was 34 °C and relative humidity (RH) was 50%.Later on, the room temperature was adjusted 23 °C and RH was 70% constantly up to the end of experiment. Lighting was maintained 24 h for first three days and 23 h from day 4 to the end of the experiment.

### 2.2. Chemical Analysis of Mushroom Stem Waste and Diets

The dried mushroom stem waste (MW) sample was ground properly and prepared (0.01 mm) for further analysis. MW samples in duplicates were analyzed for dry matter (DM), crude protein (CP), crude fiber (CF), ether extract (EE), total minerals (Ash), calcium, and phosphorus following the method of Association of Official Analytical Chemists (AOAC) [19]. Amino acids components were measured using a Hitachi L-8800 automatic amino acid analyzer (Hitachi, Tokyo, Japan). Gross energy was measure by a calorimeter (Parr 6400, Parr Instrument Company, Moline, IL 61265, USA) and the value was expressed as MJ/kg. Total phenolic content was measured by folic-ciocalteau reagent according to the method of Fu et al. [20]. In brief, a 2:1 ratio of MW extracts sample (50 µL) and folin-Ciocalteu solution (25 µL) (Merck Limited Unit, Beijing, China) were added in a microwell plate. The solution was then incubated at room temperature for 5 min. After incubation, sodium carbonate (2% NaCO_3_) (BioSino Bio-technology and Science Inc, Beijing, China) solution (25 µL), along with distilled water (100 µL), was added to the mixture. This was then further incubated at room temperature for 30 min and the absorbance was measured by Multiskan Spectrum (Thermo Fisher Scientific Instruments Company, Shanghai, China) at a wavelength of 750 nm. The value was expressed as mg of gallic acid equivalents (GAE) (Thermo Fisher Scientific Instruments Company, Shanghai, China) per gram of dry weight (mg GAE/g) basis. The active substance β-glucan content in polysaccharide was determined by a method that was previously described by Sari et al. [21]. In brief, MW was extracted with hot water and the protein content was removed with an equal content of 100 g/L tri-chloroacetic acid (Merck Limited Unit, Beijing, China). The solution was further extracted by ethanol to collect the polysaccharide precipitation. The precipitated polysaccharide was air dried by vacuum freeze drying equipment in the laboratory and the glucan cantent was measured by an assay kits (Megazyme Ltd., Beijing, China) following the manufacturing procedures. The absorbance was measured at wavelength of 510 nm. Organic matter (OM) was the calculated value from the estimated ash value (OM=100-Ash). N-free extract (NFE) was the calculated value from the equation, {NFE=dry matter-(crude protein + crude fiber + ether extract)}. The analyzed chemical composition of the experimental diets and MW were presented in Table 1 and Table 2, respectively.

### 2.3. Chicken Performance

Broiler chicken feed intake and body weight gain were recorded on a weekly basis. Total feed intake was determined as the difference between the total feed offered and total unconsumed feed by the chicks according to replicates. The waste feed that remains in the feces were separated by a wire net and measured as feed loss. Feed loss and the remaining feed in trough feeders were considered together as total unconsumed feed. Feed conversion ratio (FCR) of each replicate was then calculated as feed intake divided by body weight gain.

### 2.4. Vaccination and Serum Biochemical Analysis

All birds were immunized, with a combined vaccine of Newcastle diseases (ND, Live Strain, La Sota, Qi Lu Animal Health Care Products co. ltd, Jinan, Shandong province, China) and infectious bronchitis (IB, Live Strain, La Sota, Qi Lu Animal Health Care Products co. ltd, Jinan, Shandong province, China) on day 7 via an oculo-nasal route; infectious bursal disease vaccine (IBD, Live Strain, La Sota, Qi Lu Animal Health Care Products co. ltd, Jinan, Shandong province, China) was given on day 14 via drinking water. The vaccine booster dose was applied for the Newcastle disease vaccine (ND, Live Strain, La Sota, Qi Lu Animal Health Care Products co. ltd, Jinan, Shandong province, China) on day 22 via drinking water; infectious bursal disease vaccine (IBD, live strain, La Sota, Qi Lu Animal Health Care Products co. ltd, Jinan, Shandong province, China) was given on day 28 via drinking water. One bird was randomly selected from each replicate (sevenbirds per experimental group) and blood samples were obtained via the wing vein on day 14, day 21, day 28, and by cervical dislocation on day 42. On the respective day of blood sample collection, serum was obtained by centrifuged at 3000 × g for 20 min at 4 °C and were stored at −80 °C until measuring antibody titers, serum immune parameters (immunoglobulin A (IgA), immunoglobulin G (IgG), immunoglobulin M (IgM), interleukin-2 (IL-2), interleukin-4 (IL-4), interleukin-6 (IL-6), and tumor necrotic factor-alpha (TNF-α)) and serum metabolic profiles.Commercial enzyme-linked immune-sorbent assay (ELISA) kits (Shanghai Jianglai industrial Ltd., Shanghai, China) were used to analyze ND and IBD antibody titers.

The levels of serum immunoglobulin A (IgA), immunoglobulin G (IgG), immunoglobulin M (IgM), interleukin-2 (IL-2), interleukin-4 (IL-4), interleukin-6 (IL-6), and tumor necrotic factor-alpha (TNF-α) were done by ELISA test Kits (Shanghai Lengton Biosicences Co. Ltd., Shanghai, China)and all the absorbance was measured at the wavelength of 450 nm. The concentrations of total cholesterol (TC), triglyceride (TG), high density lipoprotein cholesterol (HDL), low-density lipoprotein (LDL), blood urea nitrogen (BUN), total protein (TP), and albumen (ALB) in the serum were measured using corresponding commercial kits (BioSino Bio-technology and Science Inc, Beijing, China) and an automatic biochemical analyzer (Hitachi 7160, Hitachi High-Technologies Corporation, Tokyo, Japan).

### 2.5. Statistical Analysis

Data were subjected to a one-way analysis of variance (ANOVA) using software SPSS for windows (SPSS Inc., Chicago, IL, USA). Significant effects of dietary treatments were evaluated with Bonferroni *t*-test to compare the means among groups. Statements of statistical significance were based on a probability of *p* < 0.05. Results were presented as the means and standard error of the means.

## 3. Results

### 3.1. Growth Performance

The effects of mushroom stem waste (MW) on growth performance of broilers are presented in Table 3. The parameters, including average daily feed intake (ADFI), average daily body weight gain (ADG), and feed conversion ratio (FCR), were not affected by dietary inclusion of MW in starter (1–21 day), finisher (22–42 day), and the overall experimental periods (1–42 day) in this study. In addition, there were no significant (*p* > 0.05) differences on initial body weight and final body weight among experimental groups in this study. However, FCR was within the standard ranges in MW fed groups in the current study.

### 3.2. Effect of Mushroom Stem Waste (MW) on Antibody Response

The effect of mushroom stem waste (MW) on antibody response is presented in Table 4. Antibody titers against Newcastle diseases (ND) were found to be higher (*p* < 0.05) in the 2% MW fed group than the control fed group (NC) and the antibiotic (PC) fed group on the both evaluating days (day 14 and day 28). On the other hand, antibody titers against infectious bursal disease vaccine (IBD) were higher (*p* < 0.05) in 2% MW fed groups than NC and PC fed groups on both evaluating days (day 21 and day 28).

### 3.3. Effect of Mushroom Stem Waste (MW) on Serum Immunity

The effect of mushroom stem waste (MW) on serum immune status was presented in Table 5. Serum immunoglobulin G (IgG) was found higher (*p* < 0.05) in both levels of MW fed groups than the control (NC) and antibiotic (PC) fed groups. Among the treatments, no significance (*p* > 0.05) differences were observed on immunoglobulin A (IgA) and immunoglobulin M (IgM) concentration in the current study. Among the serum cytokine concentration, interleukin-2 (IL-2) was higher (*p* < 0.05) in the 2% MW fed group than the NC and PC fed groups; IL-4 was higher (*p* < 0.05) in the 2% MW fed group than the NC and PC fed groups; IL-6 was higher (*p* < 0.05) in the 2% MW fed group than the NC and PC fed groups. However, tumor necrotic factor-α (TNF-α) was not affected by feeding MW in broilers in this study.

### 3.4. Effect of Mushroom Stem Waste (MW) on Serum Metabolic Profile

The effect of mushroom stem waste (MW) on serum metabolic profile was presented in Table 6. Total cholesterol (TC) concentration was lower (*p* < 0.05) in both levels of mushroom waste (MW) fed groups than that of control (NC). High density lipoprotein cholesterol (HDL) concentration was lower (*p* < 0.05) in both levels for MW fed groups compared to that of NC and PC fed groups. No significant differences were observed on low density lipoprotein cholesterol (LDL), triglyceride (TG), blood urea nitrogen (BUN), total protein (TP), and albumen (ALB) concentrations in this study.

## 4. Discussion

The information is scare regarding the effects of *F. velutipes* mushroom stem waste (MW) on performance and health status in broilers. Throughout the experimental period, the performance parameters in broilers were not affected by experimental diets, which ensured the fact that feeding mushroom waste (MW) had no any adverse effects on normal body weight gain in broilers. However, we hypothesized that this was due to the low dosages of MW in broiler diets. In addition, the higher dietary fiber in MW may affect the nutrients retention and body weight gain in experimental broilers. Similar finding were reported from our previous studies, where dietary supplementation of *F. velutipes* mushroom stem had no effects on average daily feed intake, body weight gain, and feed conversion ratio among experimental groups during the entire study period (1–70 day) in growing laying hen chickens [13].The present observations are also in agreement with previous studies by Lee et al. [22], who reported that 5% *F. velutipes* mycelium did not improve feed intake in broilers. In addition, Lee et al. [23] reported that the inclusion of fermented *F. velutipes* mycelium had no positive effect on feed intake and FCR in laying hens. No significant differences were observed on FCR fed with *Agaricus bisporus* mushroom in broilers was reported by Guimarães et al. [24] that is also justified our present findings. In contrast, dietary supplementation of *Hericiumcaput-medusae* (Bull.:Fr.) Pers. mushroom, in broiler chickens could improve body weight and FCR [25]. Giannenas et al. [26] also reported that dietary supplementation with the *Agaricus bisporus* mushroom at 2% level could improve body weight and FCR in broilers on a 42 day trial period. Higher daily body weight gain in broilers fed with *Agaricus bisporus* mushroom was also reported by Guimarães et al. [24]. Another study by Daneshmand et al. [27] reported that inclusion of oyster mushrooms, garlic, and propolis extract decreased birds’ body weight gain. This difference might be associated with mushroom species, inclusion level, and experimental bird.

Sound health can be considered from the point of view of immunology. This study hypothesized that significant higher levels of IgG, IL-2, IL-4, and IL-6 concentration in MW fed groups might be related with the presence of β-glucan in mushroom supplemented diets. A significant higher antibody response against New castle diseases (ND) and infectious bursal diseases (IBD) were noted in the current study that ensured the role of mushroom on the immune response in broilers. The purpose of applying antibiotic as positive control (PC) in this study was to compare with mushroom stem waste (MW) on antibody responses and immune globulin production in experimental broilers. Similar finding were reported from our previous studies with dietary supplementation of *F. velutipes* mushroom stem could improve the IgG, serum cytokine concentration and antibody response against ND in mushroom fed groups in pullets [13]. The mushroom polysaccharides, especially β-glucan substances, had significant immune-stimulatory functions in broiler chickens [28]. Fard et al. [29] noted that inclusion of oyster mushroom waste at 1% level could increase the serum immunity in experimental broilers. The inclusion of mushroom fed groups improved antibody response to NDV when compared to control and antibiotic diets in broilers [27]. Moreover, a higher antibody titer against ND and AI virus were noted by Toghyani et al. [30]. In contrast, no significant differences were found on the antibody titers against NDV with mushroom fed diets in broilers [31]. Similar with the study, Fard et al. [29] noted that 2% mushroom extract decreased antibody titer against ND, but could increase AI virus antibody titers and suggested to reinvestigate the antibody titer with mushroom supplement in chickens. However, no similar report was found in literature with dried *F. velutipes* mushroom waste inclusion in broiler chickens on serum immunity to compare with this study.

Another hypothesis of this study was to evaluate the lipid metabolism in broilers fed with MW. This study observed that dietary supplementation of MW had a positive role on reducing total cholesterol (TC) and high density lipoprotein cholesterol (HDL). Edible mushrooms have hypo-cholesterolemic effect and we suggest using themas oral medicine [32]. It was reported that high levels of serum TG and LDL are associated with greater risk of metabolic disorders such as fatty liver and abdominal fat deposition in chickens [33]. Our findings were in agreement with Shang et al. [34], who reported that serum total cholesterol, triglyceride, and low-density lipoprotein cholesterol levels were lower in mushroom fed groups than the control diets in broilers. In addition, dietary inclusion of mushroom (*Agaricus blazei*) powder has been reported to be reduced the total serum cholesterol concentration but no effect on serum triglyceride concentration in broilers [35]. Moreover, the total serum cholesterol was decreased in mushroom (*Pleurotus ostreatus*) supplemented diets but did not affect the other serum lipids in broilers [27]. In contrast, finding by Toghyani et al. [30] showed that serum triglyceride (TG) concentration was lower in oyster mushroom fed groups than the control in broilers. However, high-density lipoprotein, low-density lipoprotein, and total cholesterol did not differ among the treatments in their studies. It was hypothesized that higher amounts of natural fiber in mushroom waste may play a role on lowering cholesterol level by increasing lipid metabolism in chickens. Blood urea-N (BUN) was non-significantly lower in MW fed groups than the control which ensured the sound metabolic health of MW fed group broilers in the current study. Ammonia-N is the microbial product that is known to have negative health effects on birds, animals, and humans [36]. The differences on the role of lowering cholesterol with MW in this study might be related with mushroom types, inclusion levels compared to the past studies. However, hypo-cholesterolemic effect of mushrooms in poultry and poultry products is still under the reinvestigation [6].

## 5. Conclusions

Collectively this study may suggest that feeding mushroom stem waste at 2% level can be an effective way on improving the immunity in broilers as well as the effective utilization of agricultural by products to reduce environmental pollution. Further studies are needed to establish the optimum inclusion level of MW on improving performances and to detect the inner mechanisms of lipid metabolism with MW in broilers.

## Figures and Tables

**Table 1 animals-09-00692-t001:** Ingredients and nutrient composition of the experimental diets (g/kg).^1^

Items	Starter (Day 1–21)				Finisher (Day 22–42)			
	NC	PC	1% MW	2% MW	NC	PC	1% MW	2% MW
Corn	594.6	594.6	585.6	576.6	650.4	650.4	642.5	627.2
Corn gluten meal	25.0	25.0	25.0	33.0	30.0	30.0	30.0	33.0
Soybean meal	288.3	288.3	287.3	275.3	232.2	232.2	230.7	227.0
Fish meal	25.0	25.0	25.0	25.0	20.0	20.0	20.0	20.0
Soybean oil	30.0	30.0	30.0	33.0	34.0	34.0	34.0	40.0
MW ^1^	-	-	10.0	20.0	-	-	10.0	20.0
Dicalcium phosphate ^2^	15.0	15.0	15.0	15.0	10.4	10.4	10.4	10.4
Limestone	12.5	12.5	12.5	12.5	13.4	13.4	13.4	13.4
Salt	3.0	3.0	3.0	3.0	3.0	3.0	2.0	2.0
Lysine-L ^2^ (98%)	0.20	0.20	0.20	0.20	0.90	0.90	0.90	0.90
Methionine ^2^ (98%)	1.30	1.30	1.30	1.30	0.40	0.40	0.80	0.80
Threonine ^2^(98%)	0.10	0.10	0.10	0.10	0.30	0.30	0.30	0.30
Premix ^3^	5.0	5.0	5.0	5.0	5.0	5.0	5.0	5.0
Chlortetracycline	-	0.08	-	-	-	0.08	-	-
Total (g)	1000	1000	1000	1000	1000	1000	1000	1000
Chemical analysis(g/kg)								
Dry matter	880.5	876.0	877.7	880.1	879.6	879.3	874.6	876.8
Crude protein	210.1	210.1	210.0	210.3	190.2	190.1	190.3	190.2
Calcium	10.1	10.0	10.2	10.1	9.0	9.0	9.1	9.1
Phosphorus	4.6	4.5	4.4	4.4	3.5	3.4	3.4	3.5
Ether extract	56.5	56.5	56.6	59.3	61.3	61.2	61.3	67.0
Crude fiber	28.1	28.0	30.1	31.7	25.6	25.5	27.6	29.5
Calculated analysis(g/kg)								
Metabolisable energy (MJ/kg)	12.7	12.7	12.7	12.7	13.1	13.1	13.1	13.1
Lysine	11.0	11.0	11.0	11.0	10.0	10.0	9.9	10.0
Methionine	5.0	5.0	4.9	4.9	3.8	3.7	3.9	4.0
Cystine	3.2	3.2	3.1	3.2	2.8	2.7	2.8	2.9
Threonine	8.0	8.0	8.0	8.0	7.3	7.2	7.3	7.3
Tryptophan	2.8	2.8	2.8	2.8	2.4	2.4	2.4	2.4

^1^ MW = *Flammulina velutipes* mushroom stem waste; NC, negative control; PC, positive control. ^2^ Commercial available source. ^3^ Provided g/kg of the complete diet: retinyl acetate, 4500 IU; cholecalciferol, 1200 IU; DL-α-tocopheryl acetate, 2500 IU; thiamin, 5000 mg; riboflavin, 20,000 mg; phylloquinone, 10,000 mg; niacin, 45,000 mg; pantothenic acid, 35,000 mg; biotin, 1500 mg; folic acid, 3000 mg; cyanocobalamin, 40 mg; zinc, 45 mg; manganese 50 mg; iron, 30 mg; copper, 4 mg; cobalt, 120 μg; iodine, 1 mg; selenium, 120 μg.

**Table 2 animals-09-00692-t002:** Chemical compositions of *F. velutipes* mushroom stem waste (MW).^1^

Chemical Composition and Active Ingredients	Value (g/kg)
Moisture	147.37 ± 0.03
Crude protein (CP, %N × 6.25)	127.5 ± 0.66
Crude fiber (CF)	202.4 ± 0.93
Ether extract (EE)	15.0 ± 0.02
Ash (total minerals)	83.5 ± 0.04
Organic matter (OM)	916.5 ± 0.75
Nitrogen free extract (NFE)	507.73 ± 4.5
Aspartic acid	10.5 ± 0.08
Calcium	3.6 ± 0.15
Phosphorus	8.8 ± 0.23
Threonine	5.2 ± 0.08
Serine	5.1 ± 0.06
Glutamic acid	21.5 ± 0.09
Proline	5.4 ± 0.07
Glycine	5.1 ± 0.08
Alanine	7.2 ± 0.04
Valine	5.4 ± 0.03
Isoleucine	4.6 ± 0.02
Leucine	7.1 ± 0.06
Tyrosine	5.6 ± 0.07
Phenylalanine	5.5 ± 0.08
Histidine	2.3 ± 0.05
Lysine	8.1 ± 0.07
Arginine	5.6 ± 0.05
Cystine	1.3 ± 0.01
Methionine	1.4 ± 0.01
Tryptophan	1.5 ± 0.01
β-glucan	1.85 ± 0.05
Total phenolic content (mg, GAE/g)	6.56 ± 0.23
Energy (MJ/kg)	15.16 ± 0.03

^1^ Values are expressed as the mean ± SD (*n* = 3). MW = *Flammulina velutipes* mushroom stem waste. GAE: gallic acid equivalents.

**Table 3 animals-09-00692-t003:** Effect of *F. velutipes* mushroom stem waste (MW) on performance in broilers.^1^

Items	Treatments				SEM	*p*-Value
	NC (0%MW)	PC (0%MW)	1% MW	2% MW		
Average daily feed intake (g/d)						
1–21 day	37.16	38.52	37.62	36.39	0.53	0.578
22–42 day	97.28	112.63	108.96	110.93	2.40	0.093
1–42 day	67.22	75.57	73.29	73.66	1.33	0.136
Average daily body weight gain (g/d)						
1–21 day	28.01	28.21	27.96	27.74	0.40	0.983
22–42 day	52.39	67.01	62.87	63.68	2.17	0.088
1–42 day	40.19	47.61	45.42	45.71	1.17	0.136
Feed conversion ratio, FCR (g/g)						
0–21 day	1.33	1.36	1.34	1.31	0.02	0.850
22–42 day	1.85	1.68	1.75	1.74	0.07	0.307
1–42 day	1.67	1.58	1.62	1.61	0.02	0.446
IBW (g)	45.02	44.62	45.19	45.23	0.18	0.681
FBW(g)	1742	2066	1977	1992	51.36	0.127

^1^ Data represented the mean value of 63 broilers per treatment. NC, negative control; PC, positive control (antibiotic); MW, *Flammulina velutipes* mushroom stem waste. IBW, initial body weight; FBW, final body weight. SEM, pooled standard error of the means; level of significant at *p* < 0.05.

**Table 4 animals-09-00692-t004:** Effect of *F. velutipes* mushroom stem waste (MW) on antibody titers in broilers.^1^

Items	Treatments				SEM	*p*-Value
	NC (0%MW)	PC (0%MW)	1% MW	2% MW		
Newcastle disease (ND) (ng/L)						
Day14	771.38 ^b^	882.63 ^ab^	922.0 ^ab^	1045.13 ^a^	36.37	0.042
Day28	813.25 ^ab^	783.88 ^b^	914.5 ^ab^	945.13 ^a^	24.12	0.028
infectious bursal diseases (IBD) (ng/L)						
Day21	15.33 ^ab^	13.12 ^b^	16.14 ^ab^	17.64 ^a^	0.61	0.046
Day28	14.06 ^ab^	12.65 ^b^	15.85 ^ab^	16.42 ^a^	0.54	0.030

^1^ Data represented the mean value of seven samples per treatment. NC, negative control; PC, positive control (antibiotic); MW, *Flammulina velutipes* mushroom stem waste. SEM, pooled standard error of the means. a,b means that values in the same row with different letters are significantly different at *p* < 0.05.

**Table 5 animals-09-00692-t005:** Effect of *F. velutipes* mushroom stem waste (MW) on serum immune parameters in broilers (day 42).^1^

Items	Treatments					*p*-Value
	NC (0%MW)	PC (0%MW)	1% MW	2% MW	SEM	
IgA (µg/mL)	79.94	79.86	86.46	89.86	3.47	0.717
IgG (µg/mL)	45.37 ^bc^	44.48 ^c^	53.43 ^ab^	57.05 ^a^	1.63	0.002
IgM (µg/mL)	19.86	19.50	21.35	22.17	0.65	0.461
IL-2 (ng/L)	310.97 ^b^	379.29 ^ab^	389.52 ^ab^	411.96 ^a^	14.05	0.043
IL-4 (ng/L)	213.37 ^b^	265.65 ^b^	288.44 ^ab^	312.75 ^a^	15.13	0.046
IL-6 (ng/L)	155.87 ^b^	167.55 ^ab^	201.89 ^ab^	218.80 ^a^	8.87	0.020
TNF-α (ng/L)	465.97	438.32	483.77	486.55	16.54	0.759

^1^ Data represented the mean value of 7 samples per treatment. NC, negative control; PC, positive control (antibiotic); MW-*Flammlina velutipes* mushroom stem waste; IgA, immunoglobulin A; IgG, immunoglobulin G; IgM, immunoglobulin M; IL-2, interleukin-2; IL-4, interleukin-4; IL-6, interleukin-6; TNF- α, tumor necrotic factor- α. SEM-pooled standard error of the means. a,b,c means that values in the same row with different letters are significantly different at *p* < 0.05.

**Table 6 animals-09-00692-t006:** Effect of *F. velutipes* mushroom stem waste (MW) on serum metabolic profile in broilers (day 42).^1^

Items	Treatments				SEM	*p*-Value
	NC (0%MW)	PC (0%MW)	1%MW	2%MW		
TC (mmol/L)	3.26 ^a^	2.81 ^b^	2.77 ^b^	2.64 ^b^	0.08	0.005
TG (mmol/L)	0.74	0.71	0.68	0.69	0.03	0.883
HDL (mmol/L)	2.76 ^a^	2.49 ^ab^	2.10 ^c^	2.20 ^bc^	0.08	0.001
LDL (mmol/L)	0.27	0.34	0.24	0.21	0.02	0.092
BUN (mmol/L)	0.51	0.48	0.45	0.47	0.03	0.859
TP (g/L)	27.0	26.0	28.0	28.3	0.75	0.746
ALB (g/L)	12.3	12.5	12.3	13.0	0.24	0.702

^1^ Data represented the mean value of sevensamples per treatment. NC, negative control; PC, positive control (antibiotic); MW, *Flammulina velutipes* mushroom stem waste; TC, total cholesterol; TG, triglyceride; HDL, high density lipoprotein cholesterol; LDL, low density lipoprotein cholesterol; BUN, blood urea nitrogen; TP, total protein; ALB, albumen. SEM, pooled standard error of the means. a,b,c means that values in the same row with different letters are significantly different at *p* < 0.05.

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
