# Peer review of "Dietary Inclusion of Mushroom (Flammulina velutipes) Stem Waste on Growth Performance, Antibody Response, Immune Status, and Serum Cholesterol in Broiler Chickens"

_animals, 2019, doi:10.3390/ani9090692_

Round 1
Reviewer 1 Report
...

Author Response
Revision Note, (List of modification) Date: 2019-09-09 (y-m-d)
Manuscript ID: animals563664.
Dear Sir
Good day. Thank you very much for your kind consideration with our submitted article and offering us the further opportunity to submit the revised manuscript. Please find here the point to point comments with necessary changes as per suggested with this file.
Materials and Methods
There is no information about sex of the birds. It should be completed.
Response: Thank you very much, It was added in the text, please check the line number 21 and 78.
Thank you.
Reviewer 2 Report
The objectives of the paper are within the scope of the journal. Materials and methods sound good as regards the experimental diets, the number of animals and experimental units, the recordings, the analysis. However, the paper presents some major fails in the statistical analysis, in the presentation of results, and in the interpretation/discussion.
In details:
The probability of the linear and quadratic component of variance has been given in tables whereas is not clear at all if NC diet or PC diet are the “0” level. Then a Tukey test has been used to compare means. Due to the design of the experiment, with the NC and the PC diet, I recommend to use a model with the diet as the main effect, without polynomial contrasts, and with means compared by the Bonferroni t-test which should be more robust than the Tukey. The presentation of results should be based on the P value of the effect of the dietary treatment and on the results of the comparison. Results extensively present not significant differences (i.e. performance). This is not correct. Only a sentence to tell that performance were not affected by the dietary treatment is sufficient. No comparison among not-significantly higher/lower values must be provided. Introduction states that mushrooms are “alternatives to antibiotics”. Indeed this is not fully correct. They are used to improve the immune response of the animals. They cannot be used as alternatives to antibiotics. This fact should be better defined in the introduction also based on available literature (there are several reviews on the use of pre/probiotics, natural products, etc. in animal feeding). Introduction is too long. Lines 44-68 can be reduced. Lines 69-81 are too long and do not clearly introduce the topic of your paper in view of the objectives stated on lines 82-86. In other words, they appear to be out of the context. Please reduce them and provide a justification for them in relation to your study. Results should be revised based on the P value of the effect of the dietary treatment and the comparisons among treatments. Then, not significant differences should not be given. Discussion is repetitive and the biological explanation of your results is often missing. lines 254-283 on effects of mushroom inclusion on performance can be easily resumed; please explain why mushroom are expected to reduce performance or to improve them. Lines 284-287 repeat results. Combine them in discussion; provide evidence of the effects of the antibiotic inclusion; explicit why you used the PC diet and what you were expecting and relate it with the use of mushroom. Lines 302-324. Do you expect a relationship between the changes in the lipid metabolism and the improved immunity of the animals and their possible improved reaction against pathogens? Please clarify this point, if it is an additional interest for the use of mushrooms. Line 320, reference of ammonia-N as a microbial product…??? At a serum level….? What about protein metabolism? Conclusions, they repeat results and do not answer directly to the objectives of paper. Can we use mushroom in replacement of antibiotic with the same results at least for the traits you evaluated? Are they useful to improve the immune reaction of the animals? Do we need further study to clarify this issue or is this study sufficient? What is still missing? lines 326-327 and lines 220-330 repeat each other. Line 331 introduces the concept of “organic” production…no previous reference to this type of production…
Minor remarks:
Line 5. Do not use acronyms in the title.
Line 49. The reference Fard et al. is not likely to provide such general conclusion. Please, revise your introduction using reviews on the topic.
Line 90. Please, provide the sex of the animals.
Line 109. Please use “3°C”.
Line 113. Please use “(0.01 mm)”.
Line 120. Please use “kg” not “Kg” here and elsewhere.
Line 120-122. Phenolic content was measured and given, but no discussion was given about its role. Similarly, no comment is available about the content of beta-glucans, was it high? Low? Consistent with previous studies?
Line 121. The paper “22” is not a methodological paper. Please provide the complete information about the analysis you performed. The same for the paper “23”.
Lines 123-124. I do not understand this sentence “Organic matter and…from the table”.
Line 124. It should be “chemical” compositions.
Line 126. Here and throughout the manuscript, please choose and use “g/kg” or “g kg-1” consistently.
Table 1. Please avoid as much as possible acronyms. DM, CP…etc. can be fully reported. There is a sufficient amount of space in the column.
Table 2. Please revise the format. Adapt the table to the page. Also, put the unit of measurement in the heading of the tables as you did in the other tables.
Lines 145-160. Here you specify the times of immunization (7, 14 and 22 d) and the times of blood sampling (14, 21 and 28 d). If 14 d was the same day for immunization and blood sampling, please specify the order of operation and how each other were expected (or not expected) to affect each other.
Line 172-172. “Replicate was defined as an experimental unit for the trial”. Please specify which is the experimental unit for the different traits.
Line 180, 185, 193. Please delete this sub-sub headings. Line 177 is sufficient. Revise presentation of results according to major comments after statistical analysis. Do not present not significant differences.
Table 3. First line with “average feed intake” has a format different from the other items.
Line 207. Please choose and use consistently the exact P value in the text and in the tables or the reference values (0.05, 0.01, 0.001).
Line 278. Please use always the % of inclusion consistently with your previous data and discussion (here it is g/kg).
Lines 284-286. Any reference for this statement? Discussion about the nutrients that could have determined a different response in animals fed MW is poor. Only carbohydrates?
Line 304-305. Are you speaking about oral medicine in humans? Heart problems were not the focus of your paper.
Author Response
Revision Note, (List of modification) Date: 2019-09-09 (y-m-d)
Manuscript ID: animals563664.
Dear Sir
Good day. Thank you very much for your kind consideration with our submitted article and offering us the further opportunity to submit the revised manuscript. Please find here the point to point comments with necessary changes as per suggested with this attached file. We do thanks to you for the critical evaluation to make the manuscript more effective for review process in Animals Journal.
Comments: (Reviewer-2)
In details: The probability of the linear and quadratic component of variance has been given in tables whereas is not clear at all if NC diet or PC diet are the “0” level. Then a Tukey test has been used to compare means. Due to the design of the experiment, with the NC and the PC diet, I recommend to use a model with the diet as the main effect, without polynomial contrasts, and with means compared by the Bonferroni t-test which should be more robust than the Tukey. The presentation of results should be based on the P value of the effect of the dietary treatment and on the results of the comparison. Results extensively present not significant differences (i.e. performance). This is not correct. Only a sentence to tell that performance were not affected by the dietary treatment is sufficient. No comparison among not-significantly higher/lower values must be provided. Introduction states that mushrooms are “alternatives to antibiotics”. Indeed this is not fully correct. They are used to improve the immune response of the animals. They cannot be used as alternatives to antibiotics. This fact should be better defined in the introduction also based on available literature (there are several reviews on the use of pre/probiotics, natural products, etc. in animal feeding). Introduction is too long. Lines 44-68 can be reduced. Lines 69-81 are too long and do not clearly introduce the topic of your paper in view of the objectives stated on lines 82-86. In other words, they appear to be out of the context. Please reduce them and provide a justification for them in relation to your study. Results should be revised based on the P value of the effect of the dietary treatment and the comparisons among treatments. Then, not significant differences should not be given. Discussion is repetitive and the biological explanation of your results is often missing. lines 254-283 on effects of mushroom inclusion on performance can be easily resumed; please explain why mushroom are expected to reduce performance or to improve them. Lines 284-287 repeat results. Combine them in discussion; provide evidence of the effects of the antibiotic inclusion; explicit why you used the PC diet and what you were expecting and relate it with the use of mushroom. Lines 302-324. Do you expect a relationship between the changes in the lipid metabolism and the improved immunity of the animals and their possible improved reaction against pathogens? Please clarify this point, if it is an additional interest for the use of mushrooms. Line 320, reference of ammonia-N as a microbial product…??? At a serum level….? What about protein metabolism? Conclusions, they repeat results and do not answer directly to the objectives of paper. Can we use mushroom in replacement of antibiotic with the same results at least for the traits you evaluated? Are they useful to improve the immune reaction of the animals? Do we need further study to clarify this issue or is this study sufficient? What is still missing? lines 326-327 and lines 220-330 repeat each other. Line 331 introduces the concept of “organic” production…no previous reference to this type of production…
Response: Thank you very much for your critical evaluation and suggestions. We have followed all of your advises in this revised submission.
Title: Has deleted AA from the title as per suggestion.
Abstract: has been rewritten according to results on statistical values. Please check the red marking sentences in the text.
Statistical analysis: as per your advised the means values were compared by the Bonferroni t-test and has put the P value in the text and table without linear and quadratic values. Please check the results table. Please check the red marking sentences in the text.
Results: The result portion was rewritten according to suggestion, non significant results were not stated in the revised manuscript, We have put the 0 level in NC and PC diets in the results table 3,4,5,6. Please check the red marking sentences in the text.
Introduction: has been revised and only focused on the statement related with the objectives of the study. Has deleted the non related statement to reduce the introduction content. We have included the new statement about roles of phenolic components as per suggestion. Please check the red marking sentences in the text.
Materials and Methods: Has added new information as per suggestion, please check the line number. Please check the red marking sentences in the text.
Discussion: added new statement to justify our findings, as per suggestion. Please check the red marking sentences in the text.
Conclusion: has been revised and rewritten as per suggestion, Please check the red marking sentences in the text.
Minor remarks:
Line 5. Do not use acronyms in the title.
Responses: Title: Has deleted AA from the title as per suggestion. Thank you.
Line 49. The reference Fard et al. is not likely to provide such general conclusion. Please, revise your introduction using reviews on the topic.
Responses: Thank you. We have deleted the contradictory statement from the text and rewritten the introduction part on basis of the objectives of study as per suggestion.
Line 90. Please, provide the sex of the animals.
Responses: Added as per suggestion, please check the line number 21 and 78.
Line 109. Please use “3°C”.
Responses: Corrected, please check the line number 92 and 93. We have also rewritten this part as per editorial advises. Thank you.
Line 113. Please use “(0.01 mm)”.
Responses: Corrected, please check the line number 97.
Line 120. Please use “kg” not “Kg” here and elsewhere.
Responses: Corrected in text and table, please check the line number 102.
Line 120-122. Phenolic content was measured and given, but no discussion was given about its role. Similarly, no comment is available about the content of beta-glucans, was it high? Low? Consistent with previous studies?
Responses: thank you. The information was added as per suggestion; please check the line number 53-54. The chemical component of MW was within the ranges of previous published values. However, in this study, we did not compare the chemical test values with other literature as our study mainly focused on broiler performances and health status fed with mushroom waste.
Line 121. The paper “22” is not a methodological paper. Please provide the complete information about the analysis you performed. The same for the paper “23”.
Responses: thank you very much. We have revised this portion and added methodology information as per your suggestions. please check the line number 102-118.
Lines 123-124. I do not understand this sentence “Organic matter and…from the table”.
Responses: We have rewritten it to easily understandable. please check the line number 116-118. Thank you.
Line 124. It should be “chemical” compositions.
Responses: Corrected, please check the line number 118.
Line 126. Here and throughout the manuscript, please choose and use “g/kg” or “g kg-1” consistently.
Responses: Corrected through the text.
Table 1. Please avoid as much as possible acronyms. DM, CP…etc. can be fully reported. There is a sufficient amount of space in the column.
Responses: Corrected, Please check the Table 1.
Table 2. Please revise the format. Adapt the table to the page. Also, put the unit of measurement in the heading of the tables as you did in the other tables.
Responses: Corrected, Please check the Table 2.
Lines 145-160. Here you specify the times of immunization (7, 14 and 22 d) and the times of blood sampling (14, 21 and 28 d). If 14 d was the same day for immunization and blood sampling, please specify the order of operation and how each other were expected (or not expected) to affect each other.
Responses: Thank you very much for your inquire. We did ND vaccine on d7 and d22, but test was performed on d14 and d28 (table4). So it was correct pattern.
For IBD, We did IBD vaccine on d14 and d28 (revaccine), but test was performed on d21 and d28 (table4). We did not do test on d 14 for IBD, please check the table 4 for IBD-test period.
D 14 was the immunization time for IBD only, and test was done on d21 and d28. So it was also correct pattern.
Line 172-172. “Replicate was defined as an experimental unit for the trial”. Please specify which is the experimental unit for the different traits.
Responses: We have rewritten this part as per advises. Please check the line number 162-165. However, number of observation was the experimental unit for each different trail. (n=63, for Table1; n=7 for table 4,5,6). We have mentioned it in the footnote of respective Tables.
Line 180, 185, 193. Please delete this sub-sub headings. Line 177 is sufficient. Revise presentation of results according to major comments after statistical analysis. Do not present not significant differences.
Responses: Thank you. Has been revised as per advises. Please check the line number 167-174.
Table 3. First line with “average feed intake” has a format different from the other items.
Responses: Thank you. Has been corrected. Please check the table 3.
Line 207. Please choose and use consistently the exact P value in the text and in the tables or the reference values (0.05, 0.01, 0.001).
Responses: Thank you very much. It has been revised. We have put the actual p value in Table and Ref value in text.
Line 278. Please use always the % of inclusion consistently with your previous data and discussion (here it is g/kg).
Responses: Thank you very much. It has been corrected though out in the text. Please check the line number 243.
Lines 284-286. Any reference for this statement? Discussion about the nutrients that could have determined a different response in animals fed MW is poor. Only carbohydrates?
Responses: Thank you very much. We have rewritten this portion. Please check the line number 248-255.
Line 304-305. Are you speaking about oral medicine in humans? Heart problems were not the focus of your paper.
Responses: Thank you very much. Yes, in Ref 32 we have used this ref to understand the role of mushroom in lipid metabolism in human. However in Ref 33 we have stated that the adverse effects of lipid metabolism in chickens which was associated with disorders such as fatty liver and abdominal fat deposition in chickens . Please check the line number 272-274 and Ref no. 33.
Thank you.
Round 2
Reviewer 2 Report
Thank you for revising the paper according to suggestions.